# Role of Polymer Concentration and Crosslinking Density on Release Rates of Small Molecule Drugs

**DOI:** 10.3390/ijms23084118

**Published:** 2022-04-08

**Authors:** Francesca Briggs, Daryn Browne, Prashanth Asuri

**Affiliations:** Department of Bioengineering, Santa Clara University, Santa Clara, CA 95053, USA; fbriggs@scu.edu (F.B.); dbrowne@scu.edu (D.B.)

**Keywords:** controlled drug delivery, hydrogel properties, polymer concentration, crosslinking density

## Abstract

Over the past few years, researchers have demonstrated the use of hydrogels to design drug delivery platforms that offer a variety of benefits, including but not limited to longer circulation times, reduced drug degradation, and improved targeting. Furthermore, a variety of strategies have been explored to develop stimulus-responsive hydrogels to design smart drug delivery platforms that can release drugs to specific target areas and at predetermined rates. However, only a few studies have focused on exploring how innate hydrogel properties can be optimized and modulated to tailor drug dosage and release rates. Here, we investigated the individual and combined roles of polymer concentration and crosslinking density (controlled using both chemical and nanoparticle-mediated physical crosslinking) on drug delivery rates. These experiments indicated a strong correlation between the aforementioned hydrogel properties and drug release rates. Importantly, they also revealed the existence of a saturation point in the ability to control drug release rates through a combination of chemical and physical crosslinkers. Collectively, our analyses describe how different hydrogel properties affect drug release rates and lay the foundation to develop drug delivery platforms that can be programmed to release a variety of bioactive payloads at defined rates.

## 1. Introduction

Conventional drug delivery systems are often limited by poor targeting and circulation times [1,2,3,4]. Recent research has demonstrated the use of controlled drug delivery systems based on nanoparticles, liposomes, hydrogels, and membranes, amongst others, to address these issues [1,4,5]. In principle, controlled drug delivery can control when and how bioactive molecules are made available to cells and tissues, thereby enhancing their efficacy and reducing their required dosage and toxicity [2,4]. Hydrogels, in particular, have received much attention given their low toxicity and excellent biocompatibility, tunable physical properties (e.g., porosity, crosslinking density) that can be leveraged to modulate drug loads and release rates, and their ability to stabilize and protect labile biomolecules from degradation [6,7,8,9]. Furthermore, hydrogels can be prepared with a wide variety of additional physical properties, including bioinertness, biodegradability, and bioresorbability to meet the needs of specific applications [8,10,11]. They can also be formed into any desired shape or size, providing a flexible platform to meet the requirements of various drug delivery routes and systems [6,12,13]. Finally, hydrogels can be designed to respond to various biological stimuli (both internal and external), and such stimulus-responsive hydrogels have demonstrated wide applicability for the delivery of both small molecule and macromolecule drugs [8,14,15].

Currently, the majority of the studies focus on the stimulus responsiveness of hydrogels to develop a wide variety of smart systems that exhibit tailorable drug delivery properties in response to one or more (dual-responsive) stimuli, such as pH, temperature, light, enzymes, etc. [16,17,18,19]. Fewer studies focus on manipulating and studying the role of inherent hydrogel properties (i.e., monomer concentration, crosslinking density) to tailor drug delivery amounts and rates [20,21,22,23,24,25,26]. Developing a strong understanding of how innate hydrogel properties influence their ability to deliver drugs will serve to provide an additional design feature in the development of more efficacious and responsive drug delivery platforms. In this paper, we studied the role of polymer and crosslinker concentrations on the release profiles of the cancer drug, 5-fluorouracil (5-FU), using polyacrylamide (pAAm) hydrogels as the model system. pAAm hydrogel was chosen as the model as it is well characterized, commercially available, and routinely used in a wide variety of scientific applications, including for drug delivery [27,28,29,30]. These studies and results from additional supplementary experiments (using nanoparticle-based crosslinkers and double-network hydrogels) demonstrated a strong role for both polymer concentration and crosslinking density on drug delivery rates. The differences in the drug delivery rates translated into statistically significant differences in cell viability when U-87 glioblastoma cells were exposed to the different hydrogel-5-FU formulations.

## 2. Results

### 2.1. Influence of Polymer Concentration on Drug Release Rates

First, we explored the role of polymer concentration on the release rates for different concentrations of 5-FU in pAAm hydrogels (Figure 1). These experiments clearly revealed reduced drug release rates with increasing pAAm concentration. It is important to note that the differences in drug release rates are quite significant; after 24 h only, <40% of the drug had been released from 10% of the pAAm hydrogels, whereas ca. 70% of the drug had been released from 2.5% pAAm hydrogels. Furthermore, the differences in slopes for the initial and later stages of drug release for the different hydrogel samples—9, 5.6, and 1.5 for 2.5, 5, and 10% pAAm, respectively—indicated increased burst releases for lower concentrations of the hydrogel. Previous studies have indicated that higher drug loadings often result in increased burst release [31]; we were curious, therefore, to study if the polymer concentration had an effect on the burst release at higher drug loadings. Consistent with the drug release profiles for lower loadings, we observed reduced drug release rates and burst releases at higher pAAm concentrations (Figure 1b). The differences in slopes for the initial and later stages of drug release for the higher drug loadings were 9.5, 9.0, and 2.5 for 2.5%, 5%, and 10% pAAm, respectively.

### 2.2. Influence of Crosslinking Density on Drug Release Rates

Next, we studied the effect of crosslinking density on the rate of drug release from pAAm hydrogels of varying polymer concentrations. The crosslinking density was modified by changing the concentration of the chemical crosslinker, N,N′-methylenebis-acrylamide (Bis). Irrespective of the polymer concentration, we observed a decrease in drug release rates for pAAM hydrogels crosslinked using higher concentrations of Bis (Figure 2). Previous studies have indicated that the swellability of hydrogels is an important parameter in controlling drug release rates, with lower swelling ratios leading to slower release profiles [32,33,34]. To confirm this correlation, we compared the swellability of the different hydrogel formulations (Figure 3). These results clearly confirmed a significant negative correlation between crosslinking density and swellability. Taken together, results from Figure 2 and Figure 3 support previously established hypotheses that lower drug release rates for hydrogels prepared using a higher crosslinking ratio may be due to decreased swellability.

### 2.3. Nanoparticle-Mediated Increases in Crosslinking Density Affects Drug Release Rates

To confirm the role of crosslinking density on swellability and drug release rates, we performed additional experiments using pAAm hydrogels incorporating silica nanoparticles (SiNPs). Previous investigations, including our own research, have demonstrated that nanoparticles may facilitate the formation of non-covalent or pseudo crosslinks within a hydrogel network and thereby contribute to the overall crosslinking density [35,36,37,38,39]. Figure 4 compares the drug release rates and swellability for 5% pAAm prepared using different concentrations of SiNPs. The data indicated that both drug release rates and swellability decreased with increasing nanoparticle concentration; furthermore, they exhibited similar saturation behaviors (between 2 and 3% SiNPs). To offer additional support to the data that suggested that nanoparticle-mediated increases in crosslinking density can affect drug release rates, we repeated the drug release studies over a range of Bis and SiNP concentrations (Table 1). Unsurprisingly, both Bis and SiNP concentrations played a significant role on the drug release rates. Interestingly, we observed a decreased impact of SiNPs on the drug release rates at higher Bis concentrations. Collectively, these results (i.e., data presented in Figure 4 and Table 1) suggest the existence of a saturation point in the ability to control drug release rates by modulating the crosslinking density of the hydrogel network.

### 2.4. Drug Release Rates in Double-Network Hydrogels

To further validate the aforementioned observations that suggest the ability to control drug release rates by modulating the hydrogel crosslinking density, we repeated the nanoparticle studies using double-network (DN) hydrogels. In a previous study, we demonstrated that nanoparticle-mediated enhancements in the mechanical properties of DN hydrogels were strongly dependent on the extent to which SiNPs interact with the individual networks and contribute to the overall crosslinking density of the hydrogel network [40]. We were intrigued to test if these observations may be extended to drug release, or in other words, do differences in the extent of nanoparticle-mediated increases in crosslinking within DN hydrogels also impact drug release rates. Table 2 compares the drug release rates for DN hydrogels prepared using either alginate or agarose as the second network and incorporating or not incorporating SiNPs. The data reveals two important takeaways: (i) the addition of a second polymer network led to reduced drug release rates, and (ii) reduced drug release rates for DN hydrogels prepared using SiNPs relative to neat hydrogels incorporating SiNPs. Furthermore, the observed effects on SiNPs on drug release rates were consistent with the observations made for the mechanical properties of DN hydrogels due to the addition of nanoparticles [40]. We observed reduced decreases in release rates for pAAM-agarose DN hydrogels compared to pAAM-alginate DN hydrogels. We attribute these differences in SiNP mediated effects on drug release rates (and mechanical properties) to differences in the extent to which the nanoparticles may interact with the polymer chains. While it has been reported that alginate may non-covalently associate with silica nanoparticle surface [41], there are no published works (to the best of our knowledge) demonstrating positive interactions between agarose and silica nanoparticles.

### 2.5. Differences in Drug Release Rates Translates to Differences in Cytotoxicity

Finally, we performed experiments to confirm that the differences in drug release rates were biologically relevant, i.e., differences in drug release rates from the different hydrogel formulations translate to differences in compound-mediated cell viability. For this, we exposed human U-87 glioblastoma cells cultured on tissue-culture polystyrene surfaces to pAAm hydrogels, prepared using different polymer and crosslinker (Bis) concentrations, containing 5-FU. Figure 5 compares the percentage viability for U-87 cells exposed to 5-FU released from different formulations of pAAm hydrogels. These results clearly demonstrate significant differences in cell viability and a strong correlation between differences in drug release rates (mediated by differences in either polymer or crosslinker concentrations) and cell viability. We observe lower cell death for U-87 cells exposed to hydrogel formulations prepared using higher concentrations of pAAm, i.e., conditions that led to reduced drug release rates (Figure 5a). Similarly, increased crosslinking densities (achieved by increasing the concentration of Bis) led to reduced 5-FU release rates and decreased cell death (Figure 5b).

## 3. Discussion

Research over the past few years has clearly demonstrated the advantages of controlled drug delivery over conventional delivery platforms, including increased stability and bioavailability, improved and more reliable therapeutic effects, and reduced occurrence and intensity of adverse effects [1,4,42]. Hydrogels hold enormous potential for controlled drug delivery owing to their tunable physical properties, including porosity and degradability, and ability to respond to a variety of chemical and biological stimuli [6,8,43,44,45,46]. Furthermore, they are nontoxic and biocompatible, can be engineered to deliver drugs with a range of chemical properties, and possess the ability to protect labile drugs from degradation [44,46,47,48,49]. Combined, these properties enable hydrogel-based platforms to serve as excellent candidates for the controlled delivery of various therapeutic agents, including small molecule and macromolecular drugs. Hydrogel properties may be manipulated using a range of physical and chemical strategies to tailor these properties for controlled drug delivery [21,25,50]. In this study, our primary objective was to investigate if we could leverage our current understanding of manipulating the properties of hydrogels to develop drug delivery platforms with tunable release properties. Specifically, we wished to study the role of polymer and crosslinker concentrations on drug release profiles and develop an understanding of how these variables influence the ability of hydrogels to deliver drugs in a more responsive manner.

Our results provided strong evidence in support of the influence of polymer concentration and crosslinking density on drug release rates. Although others have reported similar results (i.e., negative correlations between polymer and/or crosslinker concentrations and drug release rates), we believe a more systematic approach, as performed in this work, is warranted before these properties may be used in combination with the stimulus-responsive properties as design features to develop more efficacious and flexible drug delivery platforms. To this effect, we were also intrigued to explore the correlation between nanoparticle-mediated enhancements in mechanical properties of hydrogels and the ability of nanoparticles to influence drug release properties of hydrogels. Previous investigations have clearly indicated that the ability of nanoparticles to improve hydrogel elastic modulus stems from their ability to increase the crosslinking density of polymer networks—an attribute that has also been shown to influence drug release rates from polymers [35,36,37,38,39]. Therefore, perhaps, it is not surprising that we observe a strong correlation between nanoparticle-mediated effects on hydrogel modulus and drug release rates. Our results indicated that at higher concentrations of nanoparticles, the hydrogel nanocomposite formulations exhibited a slower rate of drug release, possibly due to the increase in crosslinking density. We believe our work will provide a foundation for future studies that aim to combine one or more properties of hydrogels to develop drug delivery platforms with orthogonal design features for manipulating and controlling the release of small molecule drugs. These future studies and models may also enable the development and use of tunable hydrogel platforms, including polymeric nanocarriers and nanogels [51,52], for the delivery of a wide range of therapeutics, including peptide, protein, and nucleic acid payloads.

## 4. Materials and Methods

### 4.1. Materials

Materials for the preparation of the hydrogels, acrylamide (AAm, 40% *w*/*v*), N,N′-methylenebisacrylamide (Bis, 2% *w*/*v*), alginic acid (sodium salt, low viscosity), ammonium persulfate (APS), N,N,N′,N′-tetramethylethylenediamine (TEMED), and calcium chloride (CaCl_2_) were purchased from Sigma Aldrich (Saint Louis, MO, USA), and agarose (low melting) was purchased from Thermo Fisher Scientific (Waltham, MA, USA), and used as received. Tris-HCl buffer (pH 7.2) was purchased from Life Technologies (Carlsbad, CA, USA) and binzil silica nanoparticle colloid solution with a mean particle size of 4 nm was a gift from AkzoNobel Pulp and Performance Chemicals Inc. (Marietta, GA, USA). 5-FU was purchased from Sigma (Saint Louis, MO, USA). Human U-87 glioblastoma cells were obtained from ATCC (Manassas, VA, USA), Dulbecco’s modified Eagle medium (DMEM) from Mediatech (Manassas, VA, USA), fetal bovine serum and penicillin-streptomycin from Invitrogen (Carlsbad, CA, USA), and sodium pyruvate, MEM non-essential amino acids and GlutaMax from Life Technologies (Carlsbad, CA, USA). WST cell proliferation assay kit was purchased from Dojindo Molecular Technologies (Rockville, MD, USA).

### 4.2. Preparation of Hydrogel Samples

All hydrogel samples for drug release studies were prepared using an acrylic mold (1.6 mm thick and 6.5 mm in radius) at room temperature as previously described [37]. The polymerization reactions were performed between parallel plates of the mold to minimize exposure to air as oxygen inhibits the free radical polymerization reaction for pAAm. For pAAm samples, the monomer (AAm), crosslinker (Bis), and 5-FU stocks were diluted to their desired concentrations in pH 7.2, 250 mM Tris–HCl buffer, followed by the addition of TEMED (0.1% of the final reaction volume) and 10% *w*/*v* APS solution (1% of the final reaction volume). Final concentration of 5-FU was 1:8 *w*/*w* drug:polymer and 1:2 *w*/*w* drug:polymer for normal and high drug loadings, respectively. DN hydrogel samples composed of pAAm and alginate were prepared by first dissolving alginate in Tris–HCl buffer at room temperature. The alginate stock solutions were then diluted to the desired concentrations and added to the pAAm/5-FU reaction mixture prior to the addition of APS and TEMED, followed by the addition of 100 mM CaCl_2_ in Tris–HCl buffer to crosslink the alginate. For DN hydrogels composed of pAAm and agarose, agarose stocks were prepared by first dissolving agarose in Tris–HCl buffer at 70 °C. The pAAm reaction mixture was then warmed to 37 °C, before the addition of agarose stock solutions at 37 °C and subsequent addition of APS and TEMED. For nanocomposite hydrogels, various amounts of silica nanoparticles were added to the reaction mixture prior to the addition of APS and TEMED (and CaCl_2_ for hydrogels made using alginate).

### 4.3. Measurement of Drug Release Rates

Drug release rates for 5-FU were determined by incubating the hydrogel-5-FU samples in pH 7.2, 100 mM Tris–HCl buffer, at 37 °C. At specific time intervals, the absorbance of the releasate solution was measured at 266 nm (Tecan Infinite 200 PRO, Tecan, Switzerland). The experiments were performed in triplicate, and the average drug release rates were calculated as:(1)$$\mathrm{Drug}\;\mathrm{release},\%\;=\frac{\mathrm{Drug}\;\mathrm{released}\;\mathrm{from}\;\mathrm{the}\;\mathrm{hydrogel}}{\mathrm{Total}\;\mathrm{drug}\;\mathrm{in}\;\mathrm{the}\;\mathrm{hydrogel}}\times 100$$

### 4.4. Measurement of Swelling Rates

To measure the swelling properties of the hydrogel samples, the hydrogel disks containing no 5-FU were prepared as described above, wiped with tissue paper to remove any excess water, before weighing to determine the initial weight (W_0h_). The samples were then immersed in pH 7.2, 100 mM Tris–HCl buffer for 24 h at 37 °C. Their final weights were recorded (W_24h_) after first blotting excess buffer with tissue paper. The experiments were performed in triplicate, and the average swelling ratios were calculated as:(2)$$\mathrm{Swelling}\;\mathrm{ratio},\%\;=\frac{W_{24h}-W_{0h}}{W_{0h}}\times 100$$

### 4.5. Cell Maintenance and Measurement of Cytotoxic Responses

U-87 cells were maintained in DMEM supplemented with 15% fetal bovine serum, sodium pyruvate, MEM non-essential amino acids, GlutaMax, and 1% penicillin–streptomycin at 37 °C in a 5% CO_2_ humidified environment. Cell culture media was changed every other day and cells were passaged every 4–5 days using 0.25% trypsin/EDTA. For cell toxicity studies, U-87 cells were seeded into 96-well flat-bottom plates at a cell density of 8000–10,000 cells/well and allowed to proliferate for 24 h before exposure to the hydrogel disks containing or not containing (control) 5-FU. Cell toxic responses to the drug released from the hydrogel disks were measured by exposing the cells to the hydrogel-5-FU samples for predetermined periods of time and quantified using the commercially available, formazan-based WST assay, as per manufacturer’s instructions. Briefly, 20 mL of the WST solution for every 100 mL of culture media was added directly to the wells (after removing the hydrogel-5-FU samples) and the cells were incubated for up to 4 h in a humidified incubator at 37 °C and 5% CO_2_. 80 mL of the WST-media solution was transferred from each well into a well of a new 96-well plate in order to avoid any background absorbance, and the absorbance was measured at 570 nm (Tecan Infinite 200 PRO, Tecan, Switzerland).

### 4.6. Statistical Analysis

Average and standard error was calculated using Microsoft Excel (v. 16.54) and the standard error was presented in the form of error bars in the graphs. 

## Figures and Tables

**Figure 1 ijms-23-04118-f001:**
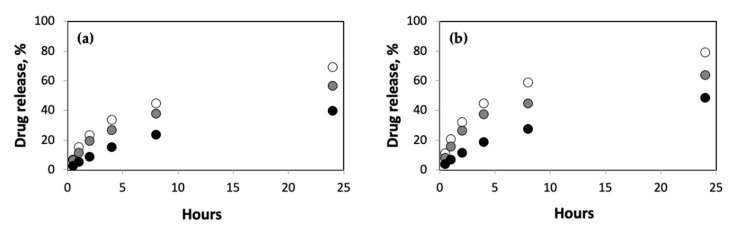
Drug release (%) from hydrogels prepared using 2.5% (white circles), 5% (grey circles), and 10% (black circles) AAm for (**a**) normal drug loading (1:8 *w*/*w* 5-FU:pAAm) and (**b**) high drug loading (1:2 *w*/*w* 5-FU:pAAm). Data shown are the mean of triplicate measurements with a standard error of <15%.

**Figure 2 ijms-23-04118-f002:**
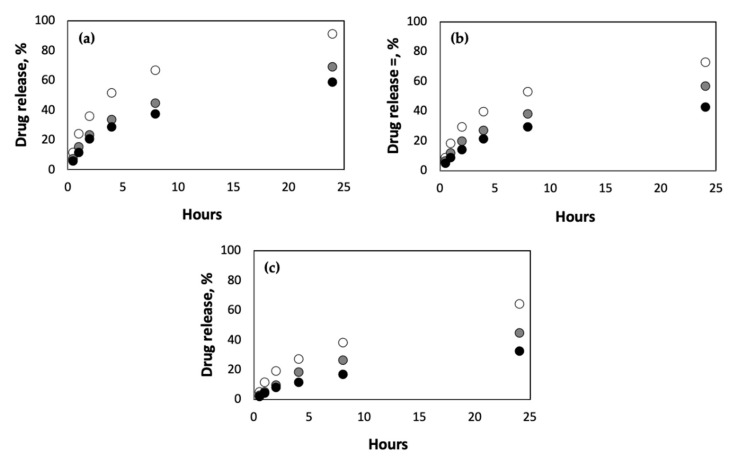
Drug release (%) from hydrogels prepared using (**a**) 2.5% AAm, (**b**) 5% AAm, and (**c**) 10% AAm, and different concentrations of Bis: 0.0625% (white circles), 0.25% (grey circles) and 1% (black circles). Data shown are the mean of triplicate measurements with standard error of <15%.

**Figure 3 ijms-23-04118-f003:**
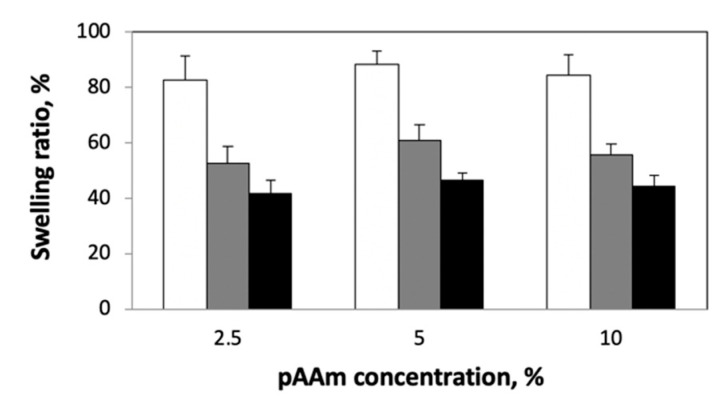
Swelling ratios (%) of hydrogels after 24 h incubation in pH 7.2 buffer and at 37 °C, prepared using different concentrations of AAm Bis: 0.0625% (white), 0.25% (grey) and 1% (black). Data shown are the mean and standard error of triplicate measurements.

**Figure 4 ijms-23-04118-f004:**
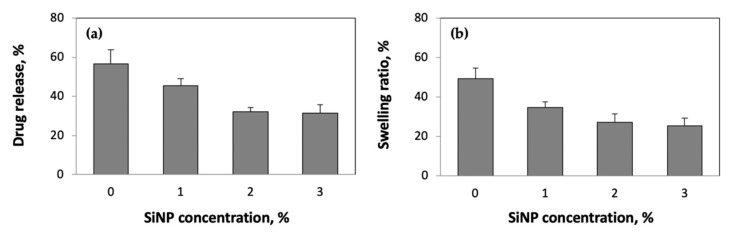
(**a**) Drug release (%) and (**b**) swelling ratios (%) for 5% pAAm hydrogels prepared using different concentrations of SiNPs after 24 h incubation in pH 7.2 buffer and at 37 °C. Data shown are the mean and standard error of triplicate measurements.

**Figure 5 ijms-23-04118-f005:**
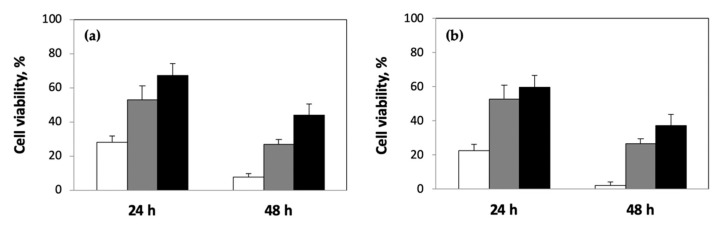
Viability of U-87 cells (%) exposed to 5-FU released from pAAm hydrogels prepared using (**a**) different concentrations of AAm: 2.5% (white), 5% (grey), and 10% (black) and 0.25% Bis, or (**b**) 5% AAm and different concentrations of Bis: 0.0625% (white), 0.25% (grey) and 1% (black). Data shown are the mean and standard error of triplicate measurements.

**Table 1 ijms-23-04118-t001:** Drug release (%) from 5% pAAm hydrogels prepared using different concentrations of Bis and SiNPs after 24 h incubation in pH 7.2 buffer and at 37 °C.

Bis, %	0% SiNPs	2% SiNPs
0.0625	76.1 ± 9	35.4 ± 3
0.25	56.6 ± 5	31.2 ± 3
1.0	42.3 ± 5	30.3 ± 2

**Table 2 ijms-23-04118-t002:** Drug release (%) from double-network hydrogels prepared using different concentrations of SiNPs after 24 h incubation in pH 7.2 buffer and at 37 °C.

Sample	0% SiNPs	2% SiNPs
5% pAAm	56.6 ± 5	31.2 ± 3
5% pAAm–1% alginate	43.2 ± 3	9.5 ± 1
5% pAAm–1% agarose	40.4 ± 5	30.3 ± 4

## Data Availability

Not applicable.

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
