# Peer review of "Role of Polymer Concentration and Crosslinking Density on Release Rates of Small Molecule Drugs"

_ijms, 2022, doi:10.3390/ijms23084118_

Round 1
Reviewer 1 Report
In this paper, authors try to investigate the role of Polymer Concentration and Crosslinking Density on Release Rates of 5-Fu as small drug. the work has novelty and interesting to researcher of this field. But I have some issues about paper that after addressing these comment the paper can be accepted.
-Author should added the morphology results of hydrogels (TEM or SEM)
-for release of 5-FU, authors used uv-vis plate reader. Why don’t use HPLC? I believe HPLC is more accurate. Please explain
- please send me absorption peak of F—FU at 266 nm.
-why authors didn’t report encapsulation efficiency (EE%) of 5-FU in hydrogel?
-I suggest authors discuses following related papers to enrich their article: https://doi.org/10.3390/pharmaceutics13030313, https://doi.org/10.1016/j.ajps.2020.05.003, https://doi.org/10.1038/s41598-021-02071-y, https://doi.org/10.1016/j.jallcom.2021.158723 and https://doi.org/10.3390/polym12061397
-Some abbreviations were used without explanation, please consider
Author Response
Thanks for your detailed comments and taking the time to improve our submission. We have addressed the comments as described below:
- Author should added the morphology results of hydrogels (TEM or SEM)
- Unfortunately, we do not have access to an TEM or SEM that can provide images of hydrogels, or other aqueous samples. Furthermore, we believe that imaging is not entirely relevant to the main results of the study. We believe that the conclusions can be supported by the analysis of swelling rates which have been included in the manuscript.
- for release of 5-FU, authors used uv-vis plate reader. Why don’t use HPLC? I believe HPLC is more accurate. Please explain
- We agree that HPLC is a more accurate methodology. However, in this work, we are reporting trends and relative values, which can be supported by plate reader experiments as reported by other studies. Moreover, it is not cost-effective to run several samples (>100 samples) as are typically needed for a drug release study using HPLC.
- please send me absorption peak of F—FU at 266 nm.
- We did not run the adsorption isotherms ourselves; instead we relied on existing literature research that have reported the peak at ~266 nm. (please see below for some examples)
- Sonochemical Development of Magnetic Nanoporous Therapeutic Systems as Carriers for 5-Fluorouracil; DOI:10.12974/2311-8792.2013.01.01.4
- A novel 5-FU/rGO/Bce hybrid hydrogel shell on a tumor cell: one-step synthesis and synergistic chemo/photo-thermal/photodynamic effect; DOI:10.1039/C6RA25834D
- We did not run the adsorption isotherms ourselves; instead we relied on existing literature research that have reported the peak at ~266 nm. (please see below for some examples)
- why authors didn’t report encapsulation efficiency (EE%) of 5-FU in hydrogel?
- Encapsulation is typically measured for samples prepared by soaking the hydrogels (or similar preparations) in the drug solutions. In our case, we included the drug in the hydrogels by adding the drug solutions to the hydrogel stocks before the formation of the samples. Or in other words, given the methodology, the EE would be ~100% for all samples.
- I suggest authors discuses following related papers to enrich their article: https://doi.org/10.3390/pharmaceutics13030313, https://doi.org/10.1016/j.ajps.2020.05.003, https://doi.org/10.1038/s41598-021-02071-y, https://doi.org/10.1016/j.jallcom.2021.158723 and https://doi.org/10.3390/polym12061397
- Thank you for the suggested references. A couple of them were not relevant, but others were added to the revised manuscript.
- Some abbreviations were used without explanation, please consider
- Thank you for the comment. We reviewed the manuscript and added the abbreviations in the revised manuscript.
Reviewer 2 Report
The authors presented a work on small things of hydrogel. Although not concept distinct, this work is meaningful. The results and presentation are clear.
- Could you provide more deep explanation of mechanism on crosslinking density-dependent release behavior (https://doi.org/10.1002/anie.202004180)?
- Is it possible to extend your findings to nanogel?
Author Response
- Could you provide more deep explanation of mechanism on crosslinking density-dependent release behavior (https://doi.org/10.1002/anie.202004180)?
- The crosslinking density dependent release can be explained by the swellability results as described in lines 90-97. To further confirm the role of swellability on drug release, we performed the nanoparticle experiments (as SiNPs can contribute to crosslinking density of pAAm hydrogels). We hope this explanation suitably explains the role of crosslinkind on drug release.
- Is it possible to extend your findings to nanogel?
- Yes, we believe so. Thank you for the comment. We added a brief edit to the discussion section (last sentence) to suggest that our findings can be extended to nanogels.
Round 2
Reviewer 1 Report
The paper can be accepted